# Boron Nanoparticle-Enhanced Proton Therapy for Cancer Treatment

**DOI:** 10.3390/nano13152167

**Published:** 2023-07-26

**Authors:** Irina N. Zavestovskaya, Anton L. Popov, Danil D. Kolmanovich, Gleb V. Tikhonowski, Andrei I. Pastukhov, Maxim S. Savinov, Pavel V. Shakhov, Julia S. Babkova, Anton A. Popov, Ivan V. Zelepukin, Maria S. Grigoryeva, Alexander E. Shemyakov, Sergey M. Klimentov, Vladimir A. Ryabov, Paras N. Prasad, Sergey M. Deyev, Andrei V. Kabashin

**Affiliations:** 1P. N. Lebedev Physical Institute of the Russian Academy of Sciences, Leninsky Prospect 53, 119991 Moscow, Russia; a.popov@lebedev.ru (A.L.P.); kdd100996@mail.ru (D.D.K.); grigorevams@lebedev.ru (M.S.G.); shemyakovae@lebedev.ru (A.E.S.); ryabov@lebedev.ru (V.A.R.); 2Bionanophotonics Laboratory, Institute of Engineering Physics for Biomedicine (PhysBio), National Research Nuclear University MEPhI (Moscow Engineering Physics Institute), Kashirskoe Shosse 31, 115409 Moscow, Russia; gvtikhonovskii@mephi.ru (G.V.T.); mssavinov@mephi.ru (M.S.S.); pvshakhov@mephi.ru (P.V.S.); babkovaserg@gmail.com (J.S.B.); aapopov1@mephi.ru (A.A.P.); smklimentov@mephi.ru (S.M.K.); pnprasad@buffalo.edu (P.N.P.); biomem@mail.ru (S.M.D.); 3Institute of Theoretical and Experimental Biophysics, Russian Academy of Sciences, 3 Institutskaya St., 142290 Pushchino, Russia; 4LP3, Aix-Marseille University, CNRS, 13288 Marseille, France; andrei.pastukhov@etu.univ-amu.fr; 5Shemyakin-Ovchinnikov Institute of Bioorganic Chemistry, Russian Academy of Sciences, 117997 Moscow, Russia; ivan.zelepukin@gmail.com; 6Department of Chemistry, Institute for Lasers, Photonics, and Biophotonics, The State University of New York at Buffalo, Buffalo, NY 14260, USA; 7“Biomarker” Research Laboratory, Institute of Fundamental Medicine and Biology, Kazan Federal University, 18 Kremlyovskaya St., 420008 Kazan, Russia; 8Institute of Molecular Theranostics, Sechenov First Moscow State Medical University, 119991 Moscow, Russia

**Keywords:** pulsed laser ablation in liquids, proton therapy, boron nanoparticles, proton boron capture therapy

## Abstract

Proton therapy is one of the promising radiotherapy modalities for the treatment of deep-seated and unresectable tumors, and its efficiency can further be enhanced by using boron-containing substances. Here, we explore the use of elemental boron (B) nanoparticles (NPs) as sensitizers for proton therapy enhancement. Prepared by methods of pulsed laser ablation in water, the used B NPs had a mean size of 50 nm, while a subsequent functionalization of the NPs by polyethylene glycol improved their colloidal stability in buffers. Laser-synthesized B NPs were efficiently absorbed by MNNG/Hos human osteosarcoma cells and did not demonstrate any remarkable toxicity effects up to concentrations of 100 ppm, as followed from the results of the MTT and clonogenic assay tests. Then, we assessed the efficiency of B NPs as sensitizers of cancer cell death under irradiation by a 160.5 MeV proton beam. The irradiation of MNNG/Hos cells at a dose of 3 Gy in the presence of 80 and 100 ppm of B NPs led to a 2- and 2.7-fold decrease in the number of formed cell colonies compared to control samples irradiated in the absence of NPs. The obtained data unambiguously evidenced the effect of a strong proton therapy enhancement mediated by B NPs. We also found that the proton beam irradiation of B NPs leads to the generation of reactive oxygen species (ROS), which evidences a possible involvement of the non-nuclear mechanism of cancer cell death related to oxidative stress. Offering a series of advantages, including a passive targeting option and the possibility of additional theranostic functionalities based on the intrinsic properties of B NPs (e.g., photothermal therapy or neutron boron capture therapy), the proposed concept promises a major advancement in proton beam-based cancer treatment.

## 1. Introduction

Based on the ability of highly energetic proton beams to destroy the DNA of rapidly proliferating cancer cells, proton therapy is considered one of most promising modalities of radiotherapy for cancer treatment [1,2,3]. The advantage of proton therapy over photon therapy and other radiotherapy methods consists of a much more selective deposition of energy following a depth–dose profile described by the Bragg curve, in which most of the dose is steeply confined at the end of the particle range (termed the “Bragg peak”) [4]. This effect makes possible the achievement of high-dose gradients in the tumor area while normal tissues remain nearly unaffected [1]. Despite the remarkable success of proton therapy, especially in the treatment of deeply seated, unresectable or recurrent tumors [2,3], this technique still needs additional tools to enhance the localization of proton beam action.

One of the recently emerged promising approaches to improve the efficiency of proton therapy relies on the use of boron-containing compounds. The hypothesis of boron proton capture therapy (BPCT) is similar in many respects to the already well-known concept of boron neutron capture therapy (BNCT), in which the boron-10 (^10^B) isotope is used to initiate a nuclear reaction leading to the generation of highly ionizing α-particles, which are short-living and known to initiate the unrepairable destruction of cancer cell DNA [5,6,7]. In BPCT, the boron-11 (^11^B) isotope, which is comprised of 80% naturally abundant boron, is used instead of ^10^B. In the original hypothesis, ^11^B-based compounds participate in a nuclear reaction p + ^11^B → 3α (p-B) [8], which is supposed to have a resonant maximum for proton energy of 675 keV [9]. The concept of BPCT was theoretically described by Joo-Young Jung et al. [10,11]. Cirrone, Manti et al. performed a series of experimental studies to confirm the feasibility of BPCT [12,13]. The authors used borocaptate sodium (BSH) solution with natural boron at concentrations of 40 and 80 ppm and observed a significant increase in the proton-induced cytogenetic effect on prostate cancer DU145 cells irradiated with a 62-MeV proton beam, as evidenced by clonogenic assays and the fluorescence in situ hybridization technique. The same authors later stated a higher efficiency of cancer cell death for a proton beam at the resonant energy of 675 keV compared to other energies, which was considered as the confirmation of the nuclear mechanism responsible for the therapeutic effect [14]. They also recently demonstrated an efficient therapeutic outcome in vivo in a glioblastoma model using boron phenylalanine (BPA) solutions [15]. Although several studies have not confirmed the reported high efficiency of such an approach [16,17,18] or dispute on the proposed mechanism of action related to nuclear proton boron capture reactions [18,19,20], the proposed concept of BPCT has attracted a lot of attention in recent years, promising a major advance in proton therapy. However, all current BPCT approaches use high concentrations of boron, typically higher than 20 mg per 1 g of tumor tissue, while the accumulation of boron-containing solutions in tumors is not selective and known to cause pronounced side effects.

We believe that, independently of the involved mechanism of action, the concept of boron-enhanced proton therapy could be further advanced by the employment of nanotechnology approaches [21,22] that employ boron-containing nanoparticles (NPs) instead of BSH or BPA solutions. In this case, one can profit from several key advantages that, first of all, include the possibility of a passive selective accumulation of boron in tumors, profiting from the enhanced permeability and retention (EPR) effect [23]. Indeed, when coated with polymers such as polyethylene glycol (PEG), NPs can accumulate in solid malignant tumors due to an inherent leakiness of the neovasculature of the growing tumor, leading to the appearance of large fenestrations between the endothelial cells lining the blood vessel. Consequently, 10–100 nm NPs can travel to the tumors. Alternatively, NPs can be actively targeted at tumors by specific molecules such as antibodies [24] or scaffold proteins [25,26]. Second, NPs can have a high boron content (especially in the case of pure elemental boron NPs), which renders possible its delivery to tumor cells at higher concentrations. Finally, one can expect a series of additional imaging and therapy functionalities based on the intrinsic physicochemical properties of NPs, similar to other nanomaterials (see, e.g., Refs. [27,28,29]), which can be enabled in parallel with the proton therapy channel to achieve a synergetic outcome.

One of main problems in the implementation of such a nanotechnology-based modality is related to the difficulty in the fabrication of low-size dispersed boron-containing NPs without surface contamination by toxic reagents and reaction by-products, which is typical for chemical synthesis routes [30,31]. NPs synthesized by dry fabrication methods such as laser pyrolysis typically have a wide size dispersion and tend to aggregate and precipitate during their dispersion in aqueous solutions [32]. To solve these problems during the preparation of B NPs, we recently proposed to employ the technique of femtosecond (fs) laser ablation in liquids, which was earlier successfully used in the synthesis of a variety of nanomaterials, including metal/semimetal [33,34,35] and semiconductor [36,37,38] NPs. Here, a crystalline B target submerged in deionized water was ablated by focused fs laser irradiation, resulting in the formation of aqueous solutions of colloidal NPs and boric acid as a by-product, which could be easily removed by centrifugation [39]. The advantage of this method consists of the cleanness of the synthesis and natural water dispersity of formed nanoparticles. Furthermore, we showed that laser-synthesized B NPs have a very low toxicity profile and can be used as sensitizers in photothermal therapy as an attractive additional modality, which could be enabled in parallel with the radiotherapy channel [39,40].

Here, we explore the use of PEG-coated elemental B NPs, synthesized by methods of fs laser ablation in water, as sensitizers of proton therapy enhancement in cellular models. We show that such NPs can indeed drastically enhance the therapeutic outcome, promising a radical advancement of proton therapy.

## 2. Materials and Methods

### 2.1. Synthesis of Boron Nanoparticles

Boron nanoparticle synthesis was performed by the pulsed laser ablation in liquids (PLAL) technique that was thoroughly described in our previous studies [34,41]. Briefly, a bulk crystalline boron target was fixed vertically inside a glass cuvette filled with 50 mL of deionized water. The thickness of the liquid layer between the inner glass wall and the boron target was 2 mm. The laser beam (1030 nm wavelength, 270 fs pulse duration, 20 µJ energy per pulse, 200 kHz frequency, Yb:KWG laser model TETA-10, Avesta, Russia) was focused on the surface of the target by using a F-theta lens with a 100-mm focal distance. The laser beam was moved along a spiral pattern on the target surface at a scanning velocity of 5 m/s using a galvanometric scanner. The duration of the ablation process was 60 min. The obtained colloidal solution was collected after the synthesis and centrifugated in two steps. The large (>150 nm) B NPs in the obtained solution were sedimented by centrifugation (at 13,000× *g*, 1 min) and removed. The remaining colloid was centrifugated again (at 20,400× *g,* 25 min), and then, the supernatant containing the boric acid formed during the synthesis was removed. The remaining residue was redispersed in water by ultrasonication.

### 2.2. NP Characterization

The morphology and size of the synthesized B NPs were characterized by the scanning electron microscopy (SEM) system (MAIA 3, Tescan, Czech Republic). Electron images were obtained at 20 keV accelerating voltage. Samples for the SEM measurements were prepared by dropping 2 μL of colloidal solution onto a cleaned monocrystalline Si substrate, which was subsequently dried at room temperature.

The hydrodynamic diameter and ζ-potential were measured by the dynamic light scattering technique using a Zetasizer Nano ZS device (Malvern Instruments, Malvern, UK). The mode values ± half-width of the peak of number-weighted size distributions were used for analysis. The Smoluchowski approximation was used for the ζ-potential calculation, and the measurement was performed in a 10 mM NaCl water solution.

FTIR spectra were measured from the B NP powders using a FT-801 spectrometer (Simex, Novosibirsk, Russia) in a total internal reflection geometry.

The size distribution of the synthesized B NPs was obtained by analysis of the SEM images in the ImageJ software environment using circle fit approximation. The final distribution was based on measurements of 300 B NP diameters.

### 2.3. Stabilization of Boron Nanoparticles

The laser-synthesized B NPs were coated with PEG using a modified Stober method, so 1 mg of B NPs was dispersed in 1 mL of 96% ethanol, and then, 65 μL of distilled water and 20 μL of 30% ammonia hydroxide were added. After thorough ultrasonication, 100 μL of 1 g L^−1^ 5 kDa Silane-mPEG(COOH) solution in ethanol was quickly dropped to a B NP colloid under stirring to start the reaction of silane chains hydrolysis and condensation. Next, B NPs were heated to 60 °C for 2 h and then incubated at room temperature overnight. The obtained B@PEG(COOH) NPs were washed 3 times with ethanol and with distilled water via centrifugation at 10,000× *g* for 15 min.

### 2.4. Proton Beam Irradiation

Cell irradiation was carried out on proton therapy complex “Prometheus” (JSC «Protom», Protvino, Russia). To irradiate cells at the Bragg peak, a rectangular field of homogenous proton irradiation 50 × 60 mm in size was formed. The energy of the protons in the beam was 160.5 MeV. To place a monolayer of cells in the position of the Bragg peak, a 151.8 mm thick polymethylmethacrylate (PMMA) moderator was installed in the beam path. The dose uniformity over the field was measured using a Gafchromic EBT3 (Ashland Inc., Wilmington, DE, USA) dosimetry film, and the absorbed dose was monitored using a PTW Unidos webline electrometer with a PTW PinPoint 3D TM31022 ionization chamber (PTW-Freiburg, Freiburg, Germany). For dosimetry at the Bragg peak, an additional PTW Bragg Peak TM34073 ionization chamber was used. The homogeneity was not less than 98% at the 95% isodose level. The accuracy of the absorbed dose determination was less than 10% for cells irradiated at the Bragg peak due to high gradients.

### 2.5. Cell Line

The experiments were carried out on a cell culture of human osteosarcoma cells of the MNNG/Hos line. MNNG/HOS cells were cultivated in DMEM/F-12 medium (1:1, Life Technologies, Carlsbad, CA, USA) containing 10% fetal bovine serum (FBS), 2 mM of L-glutamine, 100 U/mL of penicillin, 100 µg/mL of streptomycin, and a vitamin solution (PanEco, Moscow, Russia). The cells were cultivated at 37 °C in an atmosphere containing 5% CO_2_. As the cells grew and reached a sub-confluent state, they were treated with a 0.25% trypsin-EDTA solution and passed into new culture flacks at a ratio of 1:3. For proton beam irradiation, cells were seeded into 12.5 cm^2^ culture flasks (JetBiofil, Guangzhou, China) or 8-well plates (SPL LifeScience, Pochon, Korea). Before irradiation of the vials and slides, they were filled with culture medium not containing serum. The culture flasks were kept in a thermostat at 37 °C before proton beam irradiation.

### 2.6. Clonogenic Assay

Cells after irradiation were seeded in 6-well plates (SPL LifeScience, Korea) at a concentration of 2000 cells per well in DMEM/F12 + 10% FBS culture medium and cultured at 37 °C in an atmosphere containing 5% CO_2_. Colony formation was monitored daily using a CloneSelect Imager plate reader (Molecular Devices, San Jose, CA, USA). After the completion of colony formation in the control group (8–9 days), the cells were washed 3 times with phosphate-buffered saline (PBS) and fixed in 4% paraformaldehyde solution (Sigma Aldrich, Burlington, MA, USA) and stained with 0.1% crystal violet (PanEko, Moscow, Russia). Cell aggregates of more than 50 cells were considered as 1 colony. Colonies were counted manually using a magnifying glass.

### 2.7. Apoptosis Analysis

A quantitative analysis of apoptotic cells was carried out using fluorescence microscopy (NucView^®^ caspase-3 Kit, Biotium, Fremont, CA, USA). Twenty-four hours after irradiation, the cells were washed 3 times with PBS and stained by specific dye at a concentration of 5 µM. Caspase-3-positive cells were detected by a Bio-Rad ZOE imager (Bio-Rad, Hercules, CA, USA).

### 2.8. Reactive Oxygen Species Detection

Reactive oxygen species (ROS) were determined by using CellROX reagent (Invitrogen, Carlsbad. CA, USA). Cells were seeded on 8 slides (SPL LifeScience, Pocheon-si, Korea) at 2.5 × 10^3^ cells per cm^2^ in DMEM/F12 + 10% FBS culture medium. Cells were pretreated with B NPs (80 and 100 ppm) for 16 h. Before irradiation, cells were washed 3 times with PBS, and the medium was changed to a fresh one. Twenty-four hours after the irradiation, the culture medium was changed to 2 µM of CellROX solution. The cells were incubated for 60 min at 37 °C in a humidified atmosphere containing 5% CO_2_ in darkness. Micrographs were imaged at 200× magnification using a Bio-Rad ZOE imager (Bio-Rad, Hercules, CA, USA).

### 2.9. Analysis of Mitochondrial Membrane Potential (MMP)

The mitochondrial membrane potential (MMP) was determined by TMRE (tetramethylrhodamine ethyl ester, Thermo Fisher, Waltham, MA, USA) dye using fluorescence microscopy. TMRE accumulates in the mitochondrial membrane in a potentially dependent manner. Cells were seeded in 8-well tissue culture plates (SPL LifeScience, Pocheon-si, Korea) at a density of 2.5 × 10^3^ cells per cm^2^ and cultured in a CO_2_ incubator at 37 °C for 24 h with different concentrations of B NPs. The cells were preincubated with 1 μM TMRE in HBSS in a CO_2_ incubator at 37 °C for 30 min. Next, the cells were washed twice with HBSS and analyzed using a Bio-Rad ZOE (Bio-Rad, Hercules, CA, USA) imager at 200× magnification.

### 2.10. Inductively Coupled Plasma Mass Spectrometry (ICP-MS)

Cells (4 × 10^5^ per well) were incubated for 16 h with B NPs (40–100 ppm), and then, they were sedimented by centrifugation (1000× *g*, 5 min) and resuspended in 100 µL of distilled water. Then, cells were dissolved in a mixture of concentrated nitric and hydrochloric acids (Khimmed, Russia) under heating (80 °C, 15 min). Then, the samples were diluted to a 10% acid concentration, and the boron content was measured by ICP-MS with a NexION 2000 mass spectrometer (Perkin Elmer, Waltham, MA, USA). The device was calibrated by a series of boric acid solutions, and the ^11^B isotope was used for analysis.

### 2.11. Statistical Analysis

Data are presented as the mean ± standard deviation, unless otherwise stated. Statistical analysis was carried out using the two-tailed Student’s *t*-test for the colony assay. The significance in differences between experimental groups was assessed with the Mann–Whitney *U* test, which allows to detect differences in the value of a parameter between small sample groups. Statistical analysis was performed using the GraphPrism Program, version 8.0.1 (Dotmatics, USA).

## 3. Results and Discussion

### 3.1. Synthesis and Characterization of Nanoparticles

Laser ablation of the B target led to fast coloration of the liquid in the ablation chamber, indicating NP formation. The color was dark brown for the highly concentrated colloid. The boric acid formed in the synthesis and the large (>150 nm) B NPs were removed by the two stages of centrifugation, as described in the Materials and Methods section. SEM analysis showed that the B NPs had a spherical shape (Figure 1a). The physical size distribution of the particles was lognormal, with a mode diameter of 48 nm (Figure 1b).

### 3.2. Functionalization of Boron Nanoparticles

The hydrodynamic sizes of the laser-synthesized B NPs measured by DLS were 84 ± 29 nm in water (PDI: 0.162). Biomedical applications require the colloidal stability of NPs under physiological conditions. Our results showed that uncoated laser-synthesized B NPs aggregated in the physiologically relevant environment of PBS (100 mM, pH 7.4), which was confirmed by an increase in the hydrodynamic sizes to 386 ± 114 nm (PDI: 0.078). To improve the colloidal stability, the B NPs were coated with a hydrophilic polymer PEG with carboxyl terminated groups. PEGylation of the B NPs was performed by a hydrolysis of 5 kDa silane-PEG(COOH) in alkaline conditions. Coated B@PEG(COOH) NPs had a better colloidal stability in PBS, and their hydrodynamic sizes only slightly increased from 87 ± 32 nm in water (PDI: 0.122) to 97 ± 49 nm after 5 min of incubation in the buffer (PDI: 0.221) (Figure 1c). After PEGylation, we also observed a change in the ζ-potential of the NPs (Figure 1d) from a strong negative (−51 ± 9.7 mV) to more neutral values (−15.2 ± 8.6 mV).

The presence of PEG molecules on the particle surface was confirmed by FTIR spectroscopy (Figure 1e). The FTIR spectrum of uncoated B NPs showed bands associated with B-O deformation at 1090 cm^−1^, B-O stretching at 1540 cm^−1^, and a broad peak centered at 3400 cm^−1^, consistent with the B–OH stretching mode [42]. The presence of these bands could be explained by the partial oxidation of the NPs. After PEGylation of the NPs, the appearance of new bands was observed, which were characteristic of PEG; in particular, the peak at 2880 cm^−1^ was attributed to C-H stretching, and the peak at 1340 cm^−1^ could be attributed to asymmetric C-O-C stretching [43]. In addition, we observed a broad band in a region of 1700 cm^–1^, which could be partially attributed to C=O stretching of the free carboxyl groups in both the Si-PEG(COOH) and B@PEG(COOH) NP spectra.

### 3.3. Cytotoxicity Studies

B NPs were added to the MNNG/Hos cell culture to analyze the process of endocytosis and particle accumulation. A cytotoxicity analysis was performed after 72 h of coincubation with the B NPs (1–100 pmm) using the MTT test, an apoptosis analysis (Yo-Pro-1 dye) and the Live/Dead assay (Appendix A). The analysis showed a high biocompatibility of the PEG-coated B NPs without proton irradiation. High concentrations of B NPs (80 and 100 ppm) led to a slight decrease in the metabolic activity of the cells, according to the MTT test, but did not change the number of apoptotic or dead cells, according to fluorescence microscopy (Live/Dead assay and the detection of caspase-3). The B NPs did not have an inhibitory effect on the clonogenic activity of the MNNG/Hos cell culture in a wide range of concentrations from 1 to100 ppm (Appendix A). After 24 h of coincubation, a significant number of NPs were observed on the cell surface even after three-fold washing by centrifugation, which confirmed the uptake or high degree of adhesion of the B NPs to cell membranes (Figure 2a).

Additionally, a quantitative analysis of boron was carried out after 24-h incubation of the cells with B NPs (Figure 2b). We found that the cells preserved 44.7 and 59.3 µg/mL of boron after incubation with 80 and 100 ppm of B NPs, respectively. It is also worth noting that even a very high concentration of B NPs (100 ppm) did not lead to a change in cell morphology; the cells retained their spreading and morphology similar to actively dividing ones, which confirmed the good biocompatibility of the laser-synthesized B NPs.

### 3.4. Proton Beam Irradiation Experiments

At the first stage, we developed an experimental irradiation stand and analyzed the dosimetry and dose dependence on the clonogenic activity of the cells (Appendix A). Two (3 and 5 Gy) doses were selected based on the analysis of the dose-dependent cell death. We used two different methods to analyze the radiosensitization effects of B NPs under proton beam irradiation: a clonogenic assay and apoptosis analysis. The clonogenic assay assesses the ability of single cells to survive and reproduce to form colonies; however, the apoptosis analysis indicates the mechanism of cell death after proton beam irradiation. The analysis of the clonogenic activity (Figure 3) revealed a two-fold decrease in the number of cell colonies after their irradiation with a proton beam at the Bragg peak at a dose of 3 Gy, while an increase in the radiation dose to 5 Gy led to a three-fold decrease in the number of formed cell colonies compared to the non-irradiated control. The pretreatment of cells with B NPs at concentrations of 80 and 100 ppm significantly reduced the number of colonies at an irradiation dose of 3 Gy, while the further dose increase to 5 Gy did not improve the sensitizing effect of the B NPs.

Proton beam irradiation leads to the development of apoptosis triggered through various signaling pathways [44,45]. We analyzed the contribution of the B NPs to the growth of a number of apoptotic cells after proton irradiation using a selective fluorescent label NucView caspase-3 kit. As shown in Figure 4, proton irradiation led to the growth of apoptotic cells 24 h after exposure and increased their number to ~10% of the total cell quantity for both irradiation doses (3 and 5 Gy). The pretreatment of cells with B NPs at an 80 ppm concentration led to a significant increase in the number of apoptotic cells up to ~25–30% at the 3 Gy radiation dose. It should be noted that a higher dose of 5 Gy did not increase the number of apoptotic cells (similarly, 25–30% of apoptotic cells were observed). The inconsistency between the clonogenic assay and apoptosis analysis could be attributed to the different natures of the tests. The clonogenic analysis measures the formation of large cell colonies (here, from 50 cells) 8 days after irradiation. Caspase-3-positive cells indicate the number of apoptotic cells 1 day after irradiation. Firstly, although apoptosis is the primary pathway for cell death following exposure to high doses of ionizing radiation [46], there are several alternative mechanisms. Our results indicated that 70% of the cells in the B NP-treated group were caspase-3-negative after the 5 Gy radiation dose; however, only a small portion of these cells retained their ability to divide. Secondly, the different terms used to measure the radiosensitizing effect of the nanoparticles do not allow performing a direct comparison of these tests.

The observed biological effects of radiosensitization under proton beam irradiation using B NPs were found to manifest at much lower concentrations compared to molecular boron-containing agents, including BSH [18], metallacarboranes [47] and BPA [12], which presented a very encouraging sign confirming the efficiency of the suggested nanotechnology approach. The observed effect could be due to different molecular mechanisms. First, we cannot exclude the involvement of a nuclear mechanism related to the generation of alpha particles [10,11]. However, as follows from some early results of theoretical modeling [48] and previously published experimental works [16], the number of alpha particles generated due to the interaction of a 160 MeV proton beam with the used amount of boron-11 isotope might be insufficient to provide a pronounced sensitization (only a few alpha particles were produced for the 80 ppm concentration). Therefore, we considered alternative mechanisms that could be responsible for efficient cancer cell death. One such mechanism could be related to redox activity and the generation of a large number of ROS by B NPs under proton irradiation. We assumed that the internalized B NPs, having a large surface area and redox-active surface [49,50], facilitated the radiolysis of water under proton irradiation and launched ROS-induced cell cycle redistribution and apoptosis.

To check the feasibility of this alternative mechanism, we employed a CellROX-sensitive fluorescent label to determine the level of ROS in the cells with high accuracy at a specific time interval after irradiation. We carried out a quantitative assessment of the ROS and the redox status of the cells after proton irradiation (Figure 5).

We found that proton irradiation indeed led to a slight but significant increase in the level of intracellular ROS, which was confirmed by brighter-colored zones in the microphotographs (Figure 5a). At the same time, the pretreatment of the cells with B NPs at a 80 ppm concentration led to a sharp increase in the level of ROS at both 3 and 5 Gy. The increase in the B NP concentration up to 100 ppm resulted in a further increase in the ROS level. It is worth noting that the proton irradiation of cells with the 5 Gy dose in the presence of the B NPs in 100 ppm concentration led to a six-fold increase in the level of ROS compared to the non-irradiated control, which confirmed the high activity of ROS generation by the B NPs under proton irradiation (Figure 5b).

## 4. Discussion

Proton beam-based radiotherapy has a fundamental clinical advantage over conventional radiotherapy approaches [51]. The efficacy of therapeutic alpha- or beta-emitters depends on their delivery to the tumor tissue, associated with adverse effects on the thyroid gland, prostate gland, bone marrow and other organs, suffers from non-specific drug uptake. Among the beam irradiation approaches, photon and electron irradiation of the tumor region have a lower level of energy transfer to deep-seated tumors compared to proton therapy. Protons are heavily charged particles, which greatly slow down after their interaction with biological tissues. The maximum energy transfer rate is reached at the end of the ion path, resulting in a sharply localized dose peak (Bragg peak) with minimized radiation to the surrounding organs at risk [52]. The position of the Bragg peak depends on the energy of the particles, so it is possible to move the Bragg peak over the selected tumor volume by changing the proton beam energy. This leads to a wide selection of applications of proton therapy for the treatment of complicated tumors in proximity to vitally valuable organs, for example, head spinal cord and pancreatic tumors. Also, traditional beam therapies have other drawbacks. For example, the low efficiency of photon therapy is caused by a high level of tumor tissue hypoxia, which prevents the increase in the amount of DNA damage in cancer cells during their irradiation and contributes to their radioresistance [53]. Hadron therapy with heavier proton particles should have a higher level of DNA damage but should also be limited by the same drawbacks. Nevertheless, it is possible to increase the efficiency of proton therapy by using boron-containing nanoparticles, which can provide a sufficiently high local concentration of boron in the tumor tissue via the EPR effect and enhance the treatment with additional mechanisms, at least by ROS generation, as reported in this paper.

Boron-enhanced proton therapy is a promising approach in the treatment of radioresistant tumors, which can provide a highly conformal dose distribution and radiosensitizing effect in the area of tumor tissues. When a boron-containing drug is injected into a tumor, an increase in the energy release in the pathogenic region can be achieved due to the induction of alpha particles with high linear energy transfer as a result of the synthesis reaction p + ^11^B → 3α. Alpha particles formed as a result of the boron–proton reaction have an average path length in water of less than 10 µm, the value comparable with typical cell sizes, which determines the potential of using this method in therapies. The idea of using the boron-11 isotope in proton therapy is often called “proton capture therapy”, similar to the name of the boron neutron capture therapy (BNCT) method, which has recently entered the world clinical practice [49].

However, some authors have expressed doubts that, at these energies, it is possible to produce a sufficient number of alpha particles capable of providing a pronounced radiosensitizing effect similar to BNCT. Indeed, the mechanisms of action of both approaches have similarities. Neutron capture by the nuclei of the boron-10 isotope in neutron capture therapy leads to the formation of decay products: the poison of the lithium-7 atom and the alpha particle, which cause the release of additional energy in the pathogenic region. However, the values of the effective cross-sections of the nuclear reactions underlying these approaches differ significantly (in the case of a BNCT 3837 barn). The interaction of protons with the nuclei of the boron-11 isotope in “proton capture therapy” results in the formation of a composite nucleus ^12^C* in an excited state, which decays into an alpha particle and a beryllium ion ^8^Be and then decays into two alpha particles. This process is exoteric; as soon as the lowest bound state of the system of three alpha particles is reached, a total energy of 8.7 MeV is released in the form of kinetic energy transferred to the alpha particles.

Despite the wide discussion on this issue, there have been few experimental data obtained in vitro [9] and in vivo [15] confirming the effectiveness of such a binary technology to increase the efficiency of cancer cell death, but the molecular mechanisms of this effect are still poorly understood. To date, there have been several hypotheses about the mechanisms of radiosensitization in the proton–boron interaction system in cells. Some authors have claimed that the radiosensitizing effect is realized precisely due to nuclear reactions in the interaction of protons with boron-11 nuclei, which leads to the generation of alpha particles and the formation of chromosomal aberrations, apoptosis and a decrease in the clonogenic activity of cancer cells [12,13]. Other authors have hypothesized the presence of epithelial neutron interactions with boron-10 isotopes in such a system, since, when protons pass through the aqueous phase, a certain number of neutrons are generated that can effectively interact with the boron-10 isotope, which is also present in boron-containing compounds [50]. Meanwhile, it is worth noting that these are only theoretical calculations, and there is no experimental data confirming the hypothesis. Shtam et al. confirmed that, for human glioma cell cultures (A172 and Gl-Tr), the use of sodium mercaptododecaborate at concentrations of 80 and 160 ppm and their irradiation both in the middle of the 200 MeV beam of the distributed Bragg peak (SOBP) and at the distal end of the SOBP beam with an energy of 89.7 MeV is characterized by only a minor radio-sensitizing effect [18]. The molecular mechanisms of such an effect are still unclear and require detailed study, but the authors doubted that the nuclear mechanism was responsible for such an effect. Other authors have not revealed statistically significant differences in the effect of radiosensitization, claiming that there is no detectable effect of radiosensitization at these energies and concentrations of boron [16].

Meanwhile, it is worth noting that all published experimental works have reported a radio- sensitizing effect with molecular boron-containing compounds. In this study, we have shown, for the first time, the possibility of using B nanoparticles for proton cancer therapy. PEG-coated B nanoparticles were able to effectively accumulate in cancer cells and reduce clonogenic activity after irradiation with a proton beam at the Bragg peak. Such NPs can be additionally functionalized with targeted molecules to make possible a more efficient accumulation in tumor cells, which will increase the effectiveness of boron–proton therapy. We assume that the sensitizing effect of B NPs could be associated with the generation of alpha particles, but also be attributed to the generation of ROS and free radicals on the surfaces of the NPs, which leads to the activation of redox processes in cells, triggering apoptosis and subsequent cell death. This notion makes proton capture therapy similar to X-ray therapy, where the main toxicity origins from the generation of ROS through interactions with augmented electrons. This can lead to the reevaluation of agents used for the enhancement of proton therapy by shifting focus from boron compounds and testing other nanomaterials with high surface areas to generate ROS under irradiation. This issue, as well as the degree of ROS involvement in cancer cell death, require detailed assessments in further studies.

## 5. Conclusions

We assessed elemental PEG-coated B NPs prepared by methods of fs laser ablation in liquids as sensitizers of proton therapy enhancement. The MTT and clonogenic assay tests showed that the nanoformulations did not demonstrate any remarkable toxicity effects on MNNG/Hos human osteosarcoma cells for concentrations up to 100 ppm. Our experiments reported a strong proton therapy enhancement effect mediated by the B NPs. In particular, the irradiation of the cells by a 160.5 MeV proton beam at a dose of 3 Gy in the presence of 80 and 100 ppm of B NPs led to 2- and 2.7-fold decreases in the number of formed cell colonies compared to the control samples irradiated in the absence of NPs. Furthermore, we observed the proton-induced B nanoparticle-mediated generation of reactive oxygen species (ROS), which suggests a possible mechanism for cancer cell death alternative to a nuclear reaction. The proposed concept of B nanoparticle-enhanced proton therapy can profit from the passive targeting option and additional theranostic functionalities to advance cancer therapy.

## Figures and Tables

**Figure 1 nanomaterials-13-02167-f001:**
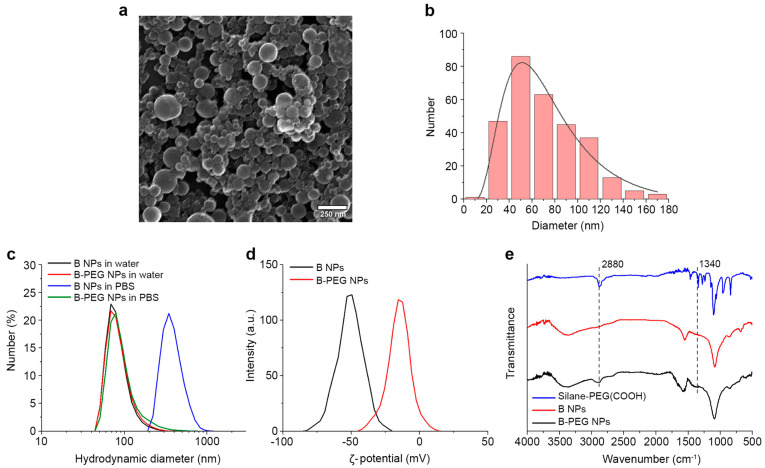
(**a**) Scanning electron microscopy image of B NPs. Scale bar-250 nm. (**b**) Physical diameter distribution of B NPs (300 particles were measured). Solid line shows lognormal fitting. (**c**) Hydrodynamic size distributions of B and B@PEG(COOH) NPs measured in water and PBS. (**d**) ζ-potential distributions of B and B@PEG(COOH) NPs. (**e**) FTIR spectra of the silane-PEG(COOH) polymer uncoated B NPs and NPs coated by the PEG(COOH) polymer.

**Figure 2 nanomaterials-13-02167-f002:**
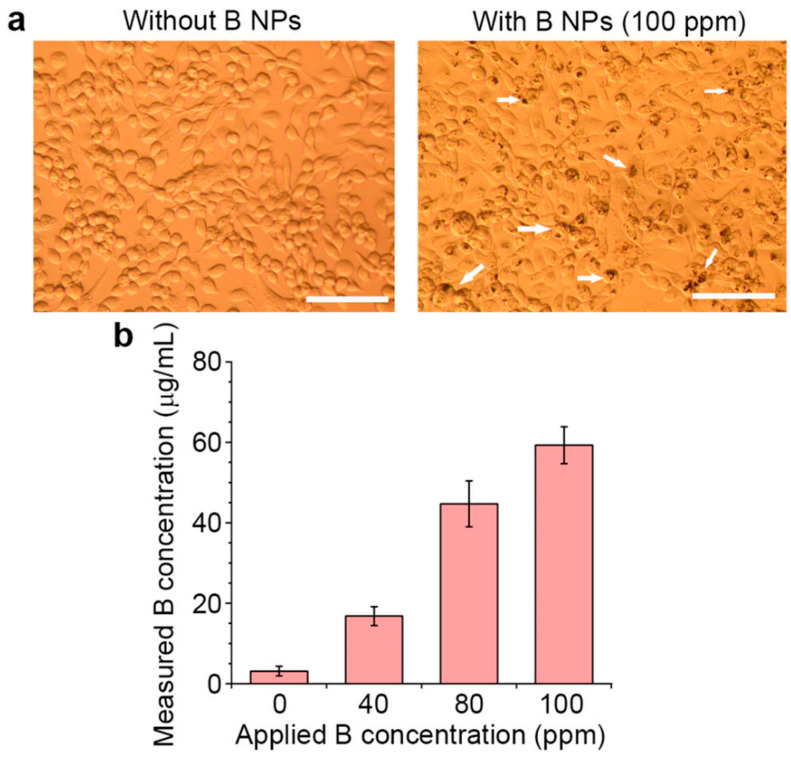
(**a**) Micrographs of MNNG/Hos osteosarcoma cells without nanoparticles and after 24 h of incubation with the B NPs at a 100 ppm concentration. Arrows indicate aggregates of B NPs on the cell surface. Scale bar—100 µm. (**b**) Quantitative analysis of boron concentration in cells by ICP-MS.

**Figure 3 nanomaterials-13-02167-f003:**
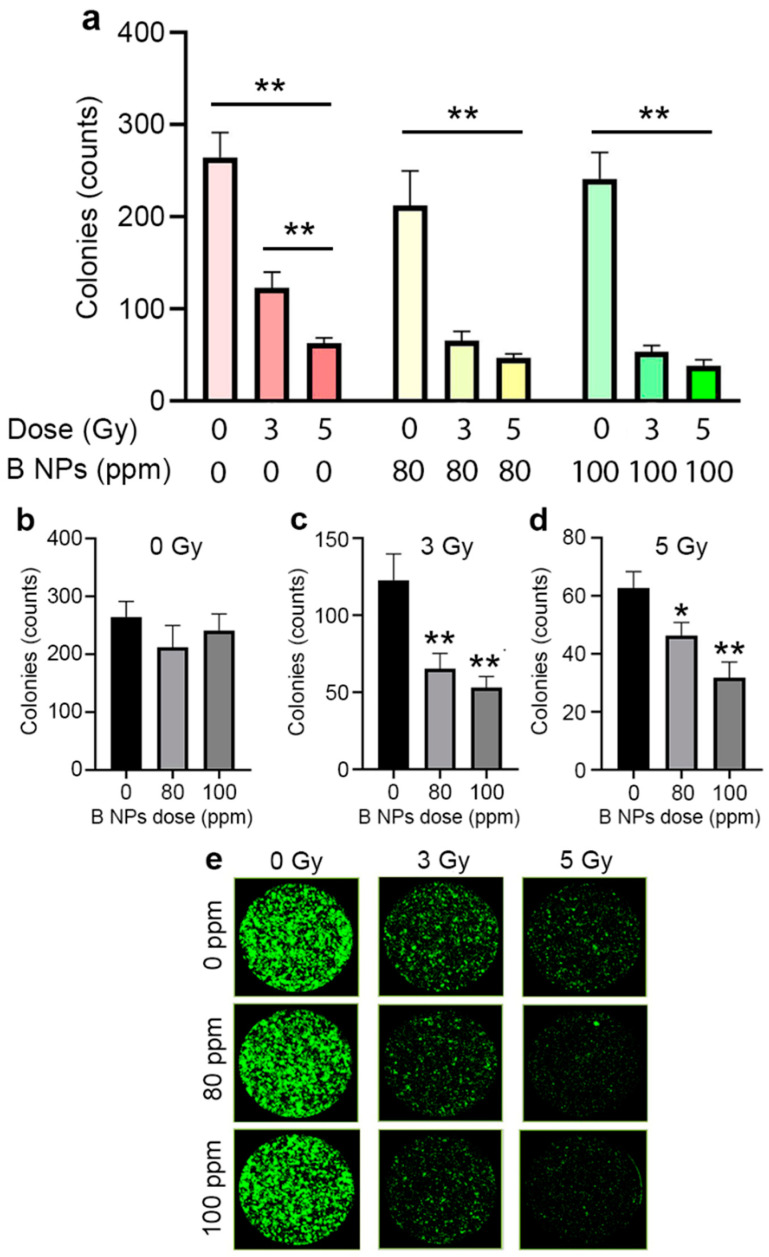
(**a**) Clonogenic analysis of the MNNG/Hos human osteosarcoma culture after incubation with the B NPs at different concentrations and their proton beam irradiation at the Bragg peak. (**b**–**d**) Dose-dependent effect of proton beam irradiation on clonogenic activity in the presence of B NPs. Cells were irradiated with 0 (**b**), 3 (**c**) and 5 (**d**) Gy doses, respectively. (**e**) Images of cell colonies obtained using a plate CloneSelect Imager reader. * *p* < 0.05; ** *p* < 0.01 via the two-tailed *t*-test.

**Figure 4 nanomaterials-13-02167-f004:**
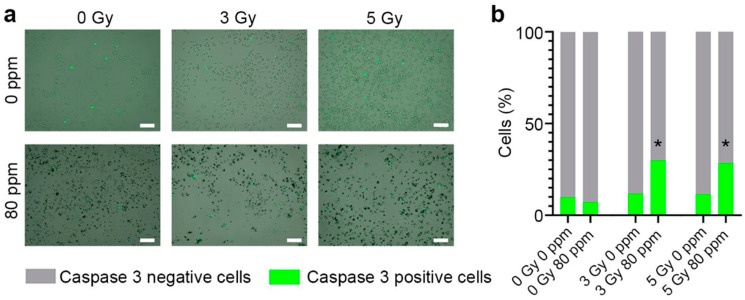
Representative images (**a**) and quantification (**b**) of MNNG/Hos cells incubated with different concentrations of B NPs and stained with NucView caspase-3 dye. Green fluorescence shows apoptotic cells. Cells were irradiated with protons at the Bragg peak at 3 and 5 Gy doses in the absence or presence of boron nanoparticles (80 ppm). Scale bar—100 µm. * *p* < 0.05 via the Mann–Whitney *U* test.

**Figure 5 nanomaterials-13-02167-f005:**
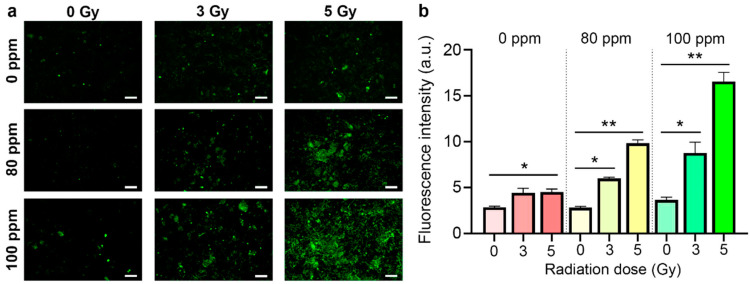
(**a**) Micrographs of MNNG/Hos osteosarcoma cells after proton beam irradiation with the B NPs (80–100 ppm). Scale bar—100 µm. (**b**) Quantitative analysis of the reactive oxygen species 24 h after proton beam irradiation in the presence of the B NPs using CellROX dye. * *p* ≤ 0.05; ** *p* < 0.01 via the Mann–Whitney *U* test.

## Data Availability

The data are available upon reasonable request from the corresponding authors.

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
