# Peer review of "Boron Nanoparticle-Enhanced Proton Therapy for Cancer Treatment"

_nanomaterials, 2023, doi:10.3390/nano13152167_

Round 1
Reviewer 1 Report
In this manuscript, the authors assessed the efficiency of B NPs as sensitizers of cancer cell kill under the irradiation. It is an interesting topic. However, the quality of the data was really low. It suggested to resubmit after a major revision.
1. Line 266, BPS (100 mM, pH 7.4) should be changed to PBS (100 mM, pH 7.4).
2. In line 272, “after X min incubation in buffer” it suggested to given the specific time.
3. Boron content in human osteosarcoma cells after 24-hour incubation was about 47 μg/mL at the concentration of 80 ppm. However, the content decreased to 40 μg/mL at a high concentration of 100 ppm. The authors should explain this.
4. In Figure 3, the colony is much lower after the irradiation at the dose of 5 Gy without B NPs as compared with that of the dose of 3 Gy. However, the caspase 3 positive cells in Figure 4b was nearly the same. How to explain this phenomenon?
5. In Figure 4a, the number of positive cells without irradiation (0 Gy) in the presence of B NPs (80 ppm) seems more than that under irradiation (3 Gy) in the presence of B NPs at the same dose. It should be explained.
6. As mentioned in table 1, Boron content in human osteosarcoma cells after 24-hour incubation was about 47 μg/mL at the concentration of 80 ppm, which was higher than that at the concentration of 100 ppm. However, the fluorescence intensity in Figure 5 was not consistent with this.
Reviewer 2 Report
In this research, the authors constructed Boron nanoparticle-enhanced proton therapy for cancer treatment. In my opinion, the current stage of this paper could meet the requirements of Nanomaterials after major revisions.
My comments are as details:
1. In table 1, the data should be shown as mean ± SD rather than the only mean data. The authors should carefully check it.
2. The advantages of using Boron nanoparticle-enhanced proton therapy rather than traditional radiotherapy (including intensive α, β, γ) should be added. Some references could be added to this part, including 10.1002/advs.202207608.
3. In all the figures, the picture scale was lost. The authors should point it out in the figures or the figure legends.
4. Some minor mistakes existed in this manuscript, such as * - p≤0.005, ** - p≤0.001 via the Mann-Whitney U-test should be * p ≤ 0.005, ** p ≤ 0.001 via the Mann-Whitney U-test.
5. How was the immune status or how immune may affect the efficacy of Boron nanoparticle-enhanced proton therapy should be discussed in the discussion. Some references could be added to this part, including 10.1016/j.ijbiomac.2022.10.167.
6. The figure legend was poorly written.
7. If possible, some in vivo bio-safety evaluation or anti-tumor efficacy may be evaluated.
Author Response
Prof. Andrei V. Kabashin
12 July 2023
Editor
Nanomaterials
Dear Editor,
Thanks for considering of our manuscript “Boron nanoparticle-enhanced proton therapy for cancer treatment” by Irina N. Zavestovskaya, et al.for a publication in Nanomaterials. We are also grateful to the reviewers for their time devoted to our manuscript, high appreciation of our work and useful remarks, which helped us to improve the presentation of results. We submitted a revision of the manuscript to take into account critical issues raised by the reviewer. Below we give our point-by-point response to Reviewer’s comments (the Reviewer’s remarks are presented in plain text, our answers follow in blue text).
We hope that you will find the revisions comprehensive enough to give the manuscript further consideration and ultimately to publish it in your journal.
Yours Sincerely,
On behalf of the authors,
Andrei V. Kabashin
CNRS Research Director,
Aix-Marseille Univ, CNRS, LP3
Marseille, France
Phone : 33-4-9182-9383
E-mail: kabashin@lp3.univ-mrs.fr
Responses to comments of Reviewer # 2
We are grateful to the Reviewer #2 for a careful evaluation of our manuscript, high appreciation of obtained results and valuable comments. Below we give our point-by-point response to Reviewer’s comments (the Reviewer’s remarks are presented in plain text, our answers follow in blue text).
REVIEWER 2
Open Review
(x) I would not like to sign my review report
( ) I would like to sign my review report
Quality of English Language
(x) I am not qualified to assess the quality of English in this paper
() English very difficult to understand/incomprehensible
() Extensive editing of English language required
() Moderate editing of English language required
() Minor editing of English language required
( ) English language fine. No issues detected
|
|
||||||
|
Yes |
Can be improved |
Must be improved |
Not applicable |
|
||
|
Does the introduction provide sufficient background and include all relevant references? |
(x) |
( ) |
( ) |
( ) |
|
|
|
Are all the cited references relevant to the research? |
(x) |
( ) |
( ) |
( ) |
|
|
|
Is the research design appropriate? |
(x) |
( ) |
( ) |
( ) |
|
|
|
Are the methods adequately described? |
(x) |
( ) |
( ) |
( ) |
|
|
|
Are the results clearly presented? |
(x) |
( ) |
( ) |
( ) |
|
|
|
Are the conclusions supported by the results? |
(x) |
( ) |
( ) |
( ) |
|
|
Comments and Suggestions for Authors
In this research, the authors constructed Boron nanoparticle-enhanced proton therapy for cancer treatment. In my opinion, the current stage of this paper could meet the requirements of Nanomaterials after major revisions.
My comments are as details:
- In table 1, the data should be shown as mean ± SD rather than the only mean data. The authors should carefully check it.
In corrected manuscript data from Table 1 was presented in Figure 2b, data are presented as mean ± SD.
- The advantages of using Boron nanoparticle-enhanced proton therapy rather than traditional radiotherapy (including intensive α, β, γ) should be added. Some references could be added to this part, including 10.1002/advs.202207608.
We have added this paragraph to the corrected manuscript in the discussion section:
“Proton beam based radiotherapy has a fundamental clinical advantage over conventional radiotherapy approaches [51]. Efficacy of therapeutic alfa- or betta-emitters depends on the delivery to the tumour tissue, and associated with adverse effects for thyroid gland, prostate gland, bone marrow and other organs, suffers from non-specific drug uptake. Among beam irradiation approaches, photons and electrons irradiation of tumour region have lower level of energy transfer to the deep-seated tumours compared to proton therapy. Protons are heavy charged particles, which greatly slow down after interaction with biological tissues. The maximum energy transfer rate is reached at the end of the ion path, resulting in a sharply localized dose peak (Bragg peak) with minimized radiation to the surrounding organs at risk [52]. The position of the Bragg peak depends on the energy of the particles, so it is possible to move the Bragg peak over the selected tumor volume by changing the proton beam energy. It led to wide application of proton therapy for the treatment of complicated tumours in proximity to vitally valuable organs, for example, head spinal cord and pancreatic tumours. Also, traditional beam therapies are associated with other drawbacks. For example, low efficiency of photon therapy was associated with the high level of tumor tissue hypoxia, which prevents an increase in the amount of DNA damage in cancer cells during their irradiation and develops their radioresistance [53]. Hadron therapy with heavier proton particles should have higher level of DNA damage, but also should be limited by the same drawback. Nevertheless, it is possible to increase the efficiency of proton therapy, by using boron-containing nanoparticles, which can provide a sufficiently high local concentration of boron in the tumor tissue by the EPR effect and enhance treatment by additional mechanisms, at least by ROS generation, as reported in this paper.…”
- In all the figures, the picture scale was lost. The authors should point it out in the figures or the figure legends.
We added the scalebars in figures 4 and 5 and indicated its size in the figure caption.
- Some minor mistakes existed in this manuscript, such as * - p≤0.005, ** - p≤0.001 via the Mann-Whitney U-test should be *p ≤ 0.005, ** p ≤ 0.001 via the Mann-Whitney U-test.
We checked manuscript and corrected this and several similar mistakes.
- How was the immune status or how immune may affect the efficacy of Boron nanoparticle-enhanced proton therapy should be discussed in the discussion. Some references could be added to this part, including 10.1016/j.ijbiomac.2022.10.167.
We kindly disagree with the Reviewer. Our study shows only in vitro effects of the boron nanoparticles on the monoculture of the cancer cells. In the absence of the immune cells, we can’t determine and discuss any immune effects of proton therapy on the observed effect. This issue is out of the scope of the presented article, so we consider the proposed discussion to be irrelevant.
- The figure legend was poorly written.
Agreed. We have made an additional description to figures 2, 4 and 5.
- If possible, some in vivo bio-safety evaluation or antitumor efficacy may be evaluated.
We appreciate this suggestion. In further studies, we will investigate the potential of proton-enhanced boron therapy for in vivo applications. This paper focuses on the in vitro effects of cancer cell damage, caused by this treatment regime and demonstrate involvement of ROS in cell damage, which is alternative mechanism of toxicity to nuclear reaction. It is impossible now to carry out in vivo experiments within the time allocated by the editor for the revision of the paper.
Reviewer 3 Report
The manuscript entitled "Boron nanoparticle-enhanced proton therapy for cancer treatment" is a very well-designed and executed study dealing with the important topic of finding new therapies for cancer treatments. I have to give praise to authors, as this is a fantastic read, where everything is very clear and makes sense, starting from a very good introduction providing good rationale of the study and ending with a great discussion. I don't really have any comments apart from the fact that it is one of best pieces of work I reviewed in recent year and it should be definitely published in Nanomaterials.
Teo very minor points:
1. In introduction there is likely a typo: PBCT instead of BPCT
2. In Table 1 in Detected concentration, µg/mL there is a lot of digits in the numbers so I would like to confirm that such high accuracy in concentration can be detected and/or what is the error of concentration detection?
I suggest a very minor revision.
Author Response
Prof. Andrei V. Kabashin
12 July 2023
Editor
Nanomaterials
Dear Editor,
Thanks for considering of our manuscript “Boron nanoparticle-enhanced proton therapy for cancer treatment” by Irina N. Zavestovskaya, et al.for a publication in Nanomaterials. We are also grateful to the reviewers for their time devoted to our manuscript, high appreciation of our work and useful remarks, which helped us to improve the presentation of results. We submitted a revision of the manuscript to take into account critical issues raised by the reviewer. Below we give our point-by-point response to Reviewer’s comments (the Reviewer’s remarks are presented in plain text, our answers follow in blue text).
We hope that you will find the revisions comprehensive enough to give the manuscript further consideration and ultimately to publish it in your journal.
Yours Sincerely,
On behalf of the authors,
Andrei V. Kabashin
CNRS Research Director,
Aix-Marseille Univ, CNRS, LP3
Marseille, France
Phone : 33-4-9182-9383
E-mail: kabashin@lp3.univ-mrs.fr
Responses to comments of Reviewer # 3
We are grateful to the Reviewer #3 for a careful evaluation of our manuscript, high appreciation of obtained results and a valuable comment. Below we give our point-by-point response to Reviewer’s comments (the Reviewer’s remarks are presented in plain text, our answers follow in blue text).
REVIEWER 3
Open Review
() I would not like to sign my review report
(x) I would like to sign my review report
Quality of English Language
() I am not qualified to assess the quality of English in this paper
() English very difficult to understand/incomprehensible
() Extensive editing of English language required
() Moderate editing of English language required
() Minor editing of English language required
(x) English language fine. No issues detected
|
Yes |
Can be improved |
Must be improved |
Not applicable |
|
|
Does the introduction provide sufficient background and include all relevant references? |
(x) |
( ) |
( ) |
( ) |
|
Are all the cited references relevant to the research? |
(x) |
( ) |
( ) |
( ) |
|
Is the research design appropriate? |
(x) |
( ) |
( ) |
( ) |
|
Are the methods adequately described? |
(x) |
( ) |
( ) |
( ) |
|
Are the results clearly presented? |
(x) |
( ) |
( ) |
( ) |
|
Are the conclusions supported by the results? |
(x) |
( ) |
( ) |
( ) |
Comments and Suggestions for Authors
The manuscript entitled "Boron nanoparticle-enhanced proton therapy for cancer treatment" is a very well-designed and executed study dealing with the important topic of finding new therapies for cancer treatments. I have to give praise to authors, as this is a fantastic read, where everything is very clear and makes sense, starting from a very good introduction providing good rationale of the study and ending with a great discussion. I don't really have any comments apart from the fact that it is one of best pieces of work I reviewed in recent year and it should be definitely published in Nanomaterials.
Teo very minor points:
- In introduction there is likely a typo: PBCT instead of BPCT
Corrected.
- In Table 1 in Detected concentration, µg/mL there is a lot of digits in the numbers so I would like to confirm that such high accuracy in concentration can be detected and/or what is the error of concentration detection?
Thank you for the suggestion. Most of these digits are invaluable in terms of high SD values. In corrected manuscript data from Table 1 was presented in Figure 2b, data are presented as mean ± SD.
I suggest a very minor revision.

Reviewer 4 Report
The paper by Zavestovskaya et al. presents a significant advancement in using the reaction
p + 11B → 3α in treatment of cancerous cells. Therefore I strongly recommend the paper for publication in Nanomaterials as it is.
The quality of English language is quite satisfactory and minor editing might be required here and there.
Author Response
Prof. Andrei V. Kabashin
12 July 2023
Editor
Nanomaterials
Dear Editor,
Thanks for considering of our manuscript “Boron nanoparticle-enhanced proton therapy for cancer treatment” by Irina N. Zavestovskaya, et al.for a publication in Nanomaterials. We are also grateful to the reviewers for their time devoted to our manuscript, high appreciation of our work and useful remarks, which helped us to improve the presentation of results. We submitted a revision of the manuscript to take into account critical issues raised by the reviewer. Below we give our point-by-point response to Reviewer’s comments (the Reviewer’s remarks are presented in plain text, our answers follow in blue text).
We hope that you will find the revisions comprehensive enough to give the manuscript further consideration and ultimately to publish it in your journal.
Yours Sincerely,
On behalf of the authors,
Andrei V. Kabashin
CNRS Research Director,
Aix-Marseille Univ, CNRS, LP3
Marseille, France
Phone : 33-4-9182-9383
E-mail: kabashin@lp3.univ-mrs.fr
Responses to comments of Reviewer # 4
We are grateful to the Reviewer #4 for a careful evaluation of our manuscript, high appreciation of obtained results and valuable comments. Below we give our point-by-point response to Reviewer’s comments (the Reviewer’s remarks are presented in plain text, our answers follow in blue text).
Open Review
() I would not like to sign my review report
(x) I would like to sign my review report
Quality of English Language
() I am not qualified to assess the quality of English in this paper
() English very difficult to understand/incomprehensible
() Extensive editing of English language required
() Moderate editing of English language required
(x) Minor editing of English language required
( ) English language fine. No issues detected
|
Yes |
Can be improved |
Must be improved |
Not applicable |
|
|
Does the introduction provide sufficient background and include all relevant references? |
(x) |
( ) |
( ) |
( ) |
|
Are all the cited references relevant to the research? |
(x) |
( ) |
( ) |
( ) |
|
Is the research design appropriate? |
(x) |
( ) |
( ) |
( ) |
|
Are the methods adequately described? |
(x) |
( ) |
( ) |
( ) |
|
Are the results clearly presented? |
(x) |
( ) |
( ) |
( ) |
|
Are the conclusions supported by the results? |
(x) |
( ) |
( ) |
( ) |
Comments and Suggestions for Authors
The paper by Zavestovskaya et al. presents a significant advancement in using the reaction
p + 11B → 3α in treatment of cancerous cells. Therefore I strongly recommend the paper for publication in Nanomaterials as it is.
Comments on the Quality of English Language
The quality of English language is quite satisfactory and minor editing might be required here and there.
We are grateful to the reviewer for the high evaluation of our work. We checked manuscript and corrected English language.

Round 2
Reviewer 1 Report
Accepted
Author Response
We are grateful to the reviewer for valuable remarks, which helped us to improve the manuscript. We are happy to know that the reviewer accepted our revised version.